# A Pharmacokinetic Study Comparing the Clearance of Vancomycin during Haemodialysis Using Medium Cut-Off Membrane (Theranova) and High-Flux Membranes (Revaclear)

**DOI:** 10.3390/toxins12050317

**Published:** 2020-05-12

**Authors:** Hussain Allawati, Linda Dallas, Sreejith Nair, Janine Palmer, Shaiju Thaikandy, Colin Hutchison

**Affiliations:** Hastings Hospital, Hawkes Bay District Health Board, Hastings 4120, New Zealand; linda.dallas@hbdhb.govt.nz (L.D.); Sreejith.Nair@hbdhb.govt.nz (S.N.); Janine.Palmer@hbdhb.govt.nz (J.P.); shaiju.thaikandy@hbdhb.govt.nz (S.T.); Colin.Hutchison@hbdhb.govt.nz (C.H.)

**Keywords:** haemodialysis, high-flux, Revaclear, Theranova, medium cut-off, vancomycin

## Abstract

Medium cut-off membrane (MCO) dialysers have been shown to remove a range of middle molecules, which are associated with adverse outcomes in haemodialysis (HD) patients, more effectively than high-flux HD. Vancomycin is widely used in HD patients for treating a variety of infections. To avoid subtherapeutic trough concentrations, it is important to understand vancomycin clearance in patients undergoing HD with the MCO membrane. This open label single centre, cross-over clinical study compared the vancomycin pharmacokinetics in chronic HD patients using MCO membrane (Theranova) and high-flux membrane (Revaclear). Five patients established on chronic HD who were due to receive vancomycin were enrolled. The study used alternating Theranova and Revaclear dialysis membranes over six consecutive sessions. Vancomycin was administered over the last one to two hours of each HD session. The maintenance dose was adjusted based on pre-HD serum concentrations. Over the 210 study samples, vancomycin clearance was higher with MCO-HD compared to high-flux HD but not statistically significant. Median percentage of vancomycin removal at 120 min by MCO membrane was 39% (20.6–51.5%) compared with 34.1% (21.3–48.4%) with high-flux HD. MCO-HD removes a slightly higher percentage of vancomycin at 120 min into dialysis compared to high-flux membrane dialysis in HD patients with infections. Application of vancomycin during the last one to two hours of each dialysis is required to maintain therapeutic concentrations to minimise loss through the dialyser and maintain therapeutic levels.

## 1. Introduction

Vancomycin is an antibiotic produced by Streptomyces orientalis, an actinomycete isolated from soil samples in Indonesia and India. Vancomycin is a glycopeptide with bactericidal action that acts by inhibiting peptidoglycan synthesis in the cell wall. It is poorly absorbed by the oral route and is excreted in large amounts in faeces. Following intravenous (IV) administration to subjects with normal kidney function, plasma protein binding is 10–55%. Eighty percent of the drug is excreted unchanged by glomerular filtration, with a half-life of 6–8 h in patients with normal glomerular function. In patients with advanced renal failure (RF), the half-life may be up to 150–250 h, and plasma protein binding is 18%. Despite its low molecular weight (1449 D), vancomycin is not removed or is minimally removed by conventional haemodialysis (HD) with low flux membranes, but its removal is increased when high permeability filters, such as polysulphone, polyacrylonitrile, or polymethylmethacrylate, are used. Monitoring vancomycin plasma levels ensures therapeutic levels (15–20 mg/L) and avoids toxic levels [1,2,3,4,5].

Vancomycin is a commonly used empirical treatment in patients with infections commonly caused by gram-positive organisms such as methicillin-resistant Staphylococcus aureus (MRSA), methicillin-sensitive Staphylococcus aureus (MSSA), coagulase-negative Staphylococci, Enterococcus faecium, or as a therapy targeted at these pathogens. These infections are frequently related to the vascular access and hence vancomycin is widely used in patients on HD. Moreover, dosing is simple according to the conventional administration regimen since the drug is infused at the end of the dialysis session and no dose is required in the period between dialysis sessions. Current guidelines and dosing recommendations are based on assuming that high flux haemodialysis removes about 20% of vancomycin per three-hour session [6,7].

A substantial number of haemodialysis patients have residual renal function which may influence vancomycin clearance. Clearance of vancomycin during haemodialysis may be further affected by membrane characteristics and dialysis prescriptions, including membrane surface area, permeability and clearance, and drug adherence to the membrane. Blood flow and dialysate flow might also influence drug removal. These conditions may lead to variability in vancomycin clearance during haemodialysis [8].

End-stage renal disease (ESRD) results in the retention of uremic toxins, which is associated with high mortality. Uremic toxins are classified into small (<500 Da) and middle-molecular weight (500 Da–60 kDa) water-soluble solutes and protein-bound substances. While conventional low-flux/high-flux hemodialysis (HD) modalities remove small solutes and smaller-sized middle-molecules, the clearance of larger middle-molecules and protein-bound substances is poor. Studies have associated middle molecules to pathological features of uremia, such as immune dysfunction and inflammation, as well as adverse outcomes in dialysis patients [9,10].

The medium cut-off (MCO) dialyser was shown to remove a wide range of middle-molecules more effectively than high-flux HD and even exceeds the performance of high-volume haemodiafiltration for large solutes, particularly Free Light Chains. This has led to the hypothesis that increased removal of these molecules will potentially reduce the risk of cardiovascular disease and secondary immune deficiency, and subsequently improve patient outcomes [11,12]. 

Currently, there is no data on the clearance of drugs by the MCO dialyser. With an increased nominal pore size along the membrane, the MCO dialyser (Theranova) and a higher sieving coefficient is anticipated to produce a drug clearance that is different on MCO compared to high-flux HD. We are postulating that there be will similar or higher clearance of vancomycin on MCO-HD when compared to using high-flux HD [11,13]. 

Our primary aim was to determine the extent of vancomycin concentration reduction and vancomycin pharmacokinetics when administered during the use of MCO (Theranova) compared with high flux dialysers (Revaclear) in chronic haemodialysis patients.

## 2. Results

In total, six patients were consented for the study. Only five patients were included as one patient was withdrawn from the study on day one of the study due to non-compliance with the study protocol. The excluded patient’s data were excluded from analysis as no data nor blood samples were collected. In this study, four patients were females (80%). Mean patient age was 65.2 years. Three patients were presumed to have residual renal function as they were on high dose loop diuretics, but the actual residual renal function was not measured. Duration of dialysis vintage was from 9 months to 4 years. All patients were dialysed three times per week, as per their usual prescribed dialysis. Dialysis access included: native fistula, graft fistula, tunneled central venous dialysis catheters, temporary central lines or a combination of different accesses. The average blood flow rate on dialysis was 287 mL/min (range 180–330). The patients included in the study and their clinical characteristics are summarised in Table 1. Dialysate flow rate was maintained at 500 mls/min in all patients irrespective of membrane used. 

The indication for treatment in each patient is summarised in Table 2. All patients experienced at least moderate infections. Bacterial organisms were aerobic gram positive: staphylococcus aureus; methicillin-resistant staphylococcus aureus (MRSA); coagulase negative staphylococcus species. One patient had cellulitis overlying the Arteriovenous Graft, which was treated as it was presumed to be a gram-positive species related infection. 

In total, 210 blood samples were obtained from five patients. With regards to protocol delivery, 100% of the samples required for the entire study were collected. All pre-HD blood samples were collected on time (100%), 60% at 5 min (median 5 min, range (5–17 min)), 46.7% at 15 min (median 17 min, range (10–35 min)), 53.3% at 30 min (median 30 min, range (30–60 min)), 56.7% at 45 min (median 45, range (40–75 min)) and 60% at 120 min (median 120 min, range (120–150 min)). Figure 1 shows delivery of protocol was similar between the two different dialysers. 

The median pre-HD vancomycin concentration was 17.2 mg/L (range 8.2–22.2). Pre-dialysis levels were subtherapeutic, less than 10 mg/L, on one occasion (3.3%). On six occasions (20%), the pre-HD concentration was more than 20 mg/L but the vancomycin dose was only withheld on four (out of 30) (13.3%). This led to dose reduction in subsequent sessions by 25%. Overall, 76.6% (23 out of 30 trough samples) maintained concentrations between 10–20 mg/L, of which 30% (9 samples) of the trough samples had concentrations less than 15 mg/L. 

Median percentage of vancomycin reduction at 120 min by MCO dialysis membrane was 39% (20.6–51.5%) compared to 34.1% (21.3–48.4%) for the high-flux dialyser. The median pre-HD concentration of vancomycin on MCO membrane dialysis declined from 15.9 mg/L (8.9–22.6 mg/L) to 9.7 mg/L (5.6–17.3 mg/L) at 120 min compared to 18.2 mg/L (11.2–21.6 mg/L) to 11.4 mg/L (7.7–15 mg/L) on high-flux membrane dialysis. Figure 2, Figure 3 and Figure 4 show that vancomycin reduction rate, concentration decline and the decrease in (log) concentration while on dialysis were higher at most time intervals with medium cut-off dialysis compared to high-flux dialysis but were not statistically significant. The 95% confidence intervals for the linear relationship as calculated by the multivariate bootstrap are also shown in Figure 4. 

These results imply (if extrapolated) that vancomycin has a half-life of 280 (95% CI 220–380) minutes on the high-flux membrane dialysis compared to 240 (95% CI 200–300) minutes on the MCO membrane dialysis (half-lives and CIs rounded to the nearest 5 min). In four patients, the median vancomycin concentration reduction at 120 min during the entire study was slightly higher with medium cut-off dialysis membrane compared to high-flux membrane (Figure 5) but was not statistically significant. Post-HD blood samples were collected with median concentration of 29.9 mg/L (11.4–39.3 mg/L). However, the true peak level would not have reached. 

Different individual patient variables (Table 3) were analysed to assess effect on percentage of vancomycin reduction at 120 min on the two different membranes. Raw analysis suggests that blood flow on dialysis (pump velocity), vancomycin trough concentration more than 15 mg/L and the use of loop diuretics resulted in a higher median percentage reduction of vancomycin at 120 min into dialysis on the MCO membrane (Table 3). 

However, a secondary analysis involved fitting a linear model to explore the effects of loop diuretics, blood flow and Body Mass Index (BMI). Too few patients were in this study to find reliable 95% confidence intervals for the model coefficients. The estimates (and their implied ratios of vancomycin concentrations, with a group not using diuretics and on the high-flux membrane dialysis) are shown in Table 4. For the BMI and blood flow estimates, this is the ratio for a 1 kg/m^2^ increase compared to the baseline (i.e., without the 1 kg/m^2^ increase) or the ratio for a 1 mL/min increase in blood flow. The results of this analysis are shown in Table 4. All effects were relatively small except for BMI, which did provide a moderate reduction in vancomycin clearance when results are scaled over a large increase (e.g., >10 points) in BMI.

We observed no Adverse Events (AEs); all volunteered, elicited and observed adverse events during the study included no reactions to delivering vancomycin in the last hour of dialysis.

## 3. Discussion

Vancomycin is an antibiotic widely used in HD units because of the high rate of infections, many of which are related to vascular access, frequently caused by gram-positive organisms. Vancomycin is the empiric treatment of choice for these patients [14,15]. This study assessed the clearance of vancomycin on MCO-HD compared to high-flux HD. 

Vancomycin reduction was 15% higher at 120 min during MCO-HD compared with high-flux HD. The vancomycin reduction ratio of 34.1% observed in this study during high-flux HD was comparable to previous studies [6,7,8]. The variability in vancomycin reduction on high-flux membrane dialysis found in our study has been described in other studies [16]. Intradialytic clearance of vancomycin varied on the different six days of the study period and when alternated between the two different types of membranes (raw data in Appendix A).

In this study, blood flow on dialysis (pump velocity), vancomycin trough concentration of more than 15 mg/L and the use of loop diuretics seemed to have some effect on vancomycin concentration at 120 min but were not statistically significant. We used loop diuretic as a surrogate for residual renal function. Pre-dialysis trough concertation, blood flow and residual renal function have been shown in other studies to affect the removal of vancomycin [17,18]. 

Our current hospital protocol recommends that only 500 mg of vancomycin be infused per hour to avoid infusion complications such as Red Man Syndrome. However, in this study vancomycin dose was administered over the last one to two hours of dialysis in our patients. Although the use of vancomycin during dialysis may lead to a certain loss of drug, this strategy is more practical, and the loss of vancomycin may be compensated to a lesser degree by administration of higher vancomycin doses based on therapeutic drug monitoring [8]. While the number of patients in this study is small, we observed no adverse events related to vancomycin infusion rate in all the thirty sessions of haemodialysis. 

Estimating the amount of drug removed from the percentage decline in plasma concentrations during dialysis assumes a single compartment model and the absence of redistribution. Neither of these assumptions is true for vancomycin. Consequently, drug removal estimated from the decline in plasma levels overestimates dialytic clearance and supplemental dosage needs [19,20]. The rebound of vancomycin serum concentrations occurs following dialysis with highly permeable membranes [21,22]. In our study we could not estimate the effect of rebound as no peak post-dialysis samples were collected, and patients received vancomycin during their dialysis. From a practical point of view, true peak samples were not obtainable. 

The existing dosing for patients undergoing intermittent dialysis may not be applicable to unwell or septic chronic dialysis patients. For example, sepsis can lead to the development of endothelial damage and increase capillary permeability, which cause displacement of fluids from the vasculature into the interstitium. The volume of distribution of some drugs in septic patients may differ substantially from that reported in pharmacokinetic studies on healthy individuals [23]. However, we and other studies have observed a similar percentage of vancomycin reduction in high flux membranes [6,7,8].

Optimal vancomycin dosing in patients who are undergoing haemodialysis is controversial. Clinicians commonly select the dose of vancomycin on the basis of actual body weight (mg/kg) in accordance with data generated from patients with normal kidney function [24]. An AUC/MIC (Area Under Curve/Minimum Inhibitory Concentration) ratio equal to or greater than 400 has been advocated as a target to achieve clinical effectiveness with vancomycin therapy. However, it can be difficult in the clinical setting to obtain multiple serum vancomycin concentrations to determine the exact AUC and subsequently calculate AUC/MIC. Trough serum concentration monitoring, which can be used as a surrogate marker for AUC, is recommended as the most accurate and practical method for vancomycin monitoring. Trough vancomycin serum concentrations maintained between 10–15 mg/L are recommended for mild infections and 15–20 mg/L in severe infections [24]. Therefore in our study, intermittent maintenance dosing in the last one to two hours of dialysis seems to achieve adequate trough concentration in the majority (about 75%) of trough samples.

The limitations of this study include collection of data from patients undergoing haemodialysis, without quantification of the degree of residual renal function. Also, our patient numbers for this study were small, limiting the analysis of patient characteristics on drug clearance. The confidence intervals for the main analysis were wide and reliable intervals for multivariable analyses were not able to be obtained. A larger study with more participants would improve the statistical analysis. Finally, subsequent studies may choose to investigate the potential wider impact of vancomycin on the immune system either directly or through altering the gut microbiome.

## 4. Conclusions

While there is a wide range of recommendations regarding dosing of vancomycin in patients on high-flux HD, this is the first study to evaluate the pharmacokinetics of vancomycin clearance on the novel MCO dialysers. We identified vancomycin removal to be higher on MCO dialysis compared with high-flux HD with a predicted shortening of serum half-life to 240 min from 280 min. The clearance of vancomycin on HD is already significantly altered by patient characteristics and the dialysis script, therefore the addition of the MCO membrane provides another variable to the equation. We therefore recommend dosing of vancomycin to be based on the pre-HD trough levels on an individual basis to avoid sub-therapeutic treatment. 

## 5. Materials and Methods 

This is an open label, single centre, non-randomised clinical study performed at a Regional Secondary Care Hospital in New Zealand. The study was conducted in accordance with the Declaration of Helsinki. The study protocol was approved by the local Hospital’s Ethics Committee, the local hospital Maori Research review committee and the Central Health and Disability Ethics Committee (Central-HDEC, Ethics ref: 18/CEN/252, approval date:18 December 2018). The study was registered with The Australian New Zealand Clinical Trial Registry (ANZCTR, Trial ID: ACTRN12618001895279). An interpreter was offered when needed and written informed consent was obtained from patients prior to enrolling in the study. All subjects gave their informed consent for inclusion before they participated in the study.

### 5.1. Subjects

Five chronic intermittent, in-centre haemodialysis patients with an indication to treat with vancomycin for a minimum of six sessions of HD (i.e., two weeks duration) were included. 

The inclusion criteria included being over the age of 18; an in-centre haemodialysis patient established on a MCO dialyser (Theranova); able to provide written informed consent and have the ability to understand the requirements of the study and to comply with the study protocol. 

The exclusion criteria was clinical evidence of any active significant disease that could potentially compromise the ability of the investigator to evaluate or interpret the effects of the study treatment on safety assessment and thus increase the risk to the subject to unacceptable levels; on other concomitant antibiotics at the time of the study; female subjects of childbearing age who had positive serum pregnancy test at baseline; lactating at the time of the study; documented allergic or adverse reaction to vancomycin previously; unable to understand and comply with the study; on Haemodiafiltration (HDF) and those who have sensitivity (allergic reaction) to Revaclear or Theranova membranes. 

### 5.2. Study Procedure

The study was conducted between January 2019 and August 2019. The subjects meeting the entry criteria received a loading dose of intravenous vancomycin at 30 mg/kg (rounded to the nearest 50 mg) to the maximum of 2000 mg administered in an intravenous infusion (IV) which was determined by their treating doctor. All subjects received vancomycin in the last 60–120 min of a dialysis session. The infusion rate varied depending on the dose administered (Table 5) [24,25,26].

The Principal Investigator then approached the subject for inclusion in the study and provided the patient information sheet and when required used a suitable interpreter from the District Health Board (DHB). Subjects had 48–72 h to consent to the study after the initial dose of vancomycin was delivered, i.e., the loading dose delivery was not part of the study. 

Once written consent was obtained from the patients wishing to participate in the study, the following data were recorded; demographics (date of birth, gender, ethnic origin); medical history (including concomitant medication use, allergies, and medical conditions); complete physical examination (including vital signs, dry weight (kg), and weight on day of study) and HD access type; blood flow rate; dialysate flow rate and dialyser surface area. 

Intermittent haemodialysis was performed in all patients with the batch dialysis system (GAMBRO, Baxter International Inc., Deerfield, IL, USA). Patients commenced the study on MCO dialysers (Theranova™, surface area 1.8 m^2^, Baxter International Inc., Deerfield, IL, USA) and alternated with high-flux dialysers (Revaclear 400™, surface area 1.8m^2^, Baxter International Inc., Deerfield, IL, USA). The duration of the study for each subject was the next six HD sessions (approximately 14 days, i.e., the 6 HD sessions following the loading dose of vancomycin). 

At each visit, blood samples for drug concentration were collected at the start of a HD session, then at 5, 15, 30, 45, and 120 min after starting dialysis, and 30 min post-dialysis. Blood samples (3–5 mL) were collected from the arterial line of a dialysis circuit in a tube (BD Vacutainer Barricor Plasma Blood Collection Tube). The samples were analysed at the local laboratory. Vancomycin concentrations were determined using in vitro chemiluminescent microparticle immunoassay (CMIA) (the Abbott Architect 8200 analyser (*i*Vancomycin), Abbott Laboratories, Diagnostics Division, Abbott Park, IL, USA).

A vancomycin pre-haemodialysis concentration of 10–20 mg/L was chosen as a target concentration for vancomycin dosage adjustment. Dose adjustments were made in an approximately linear fashion in accordance with trough plasma levels. For example, an increase in dose by 50% would result in an increase in trough levels by approximately 50% and vice versa. An approximate maintenance dose of 10–15 mg/kg was administered at each session when pre-dialysis concentration was less than or equal to 20 mg/L, depending on the indication of treatment. The dose was delivered in the last 60–120 min of a dialysis session as per Table 5. Vancomycin was reconstituted at the time of use. The 500 milligrams vials (Mylan New Zealand Ltd., Auckland, New Zealand) were reconstituted with 10 mL of water for injections and every 500 mg was diluted with 250 mL of Sodium Chloride Intravenous Infusion 0.9%. The extra intravenous fluid needed for the drug was ultrafiltrated during dialysis. The dose was administered using a controlled infusion pump into venous line of a dialysis circuit. 

Subjects remained at the dialysis unit for 30 min post-infusion and were discharged if no adverse events (AEs) were observed. Adverse events were recorded in a data sheet. Subjects returned to the unit at their next arranged haemodialysis session. 

The Principle Investigator reviewed all test results and AEs reported prior to the next dose. The dose was adjusted based on a review of pre-dialysis vancomycin trough concentration by the Principle investigator. 

### 5.3. Statistics

All statistical analyses were performed in R version 3.5.1 ‘Feather Spray’ (R Core Team (2018). R: A language and environment for statistical computing. R Foundation for Statistical Computing, Vienna, Austria, https://www.R-project.org/). Some data were analysed in MS Excel (Office 365, Version 1908 (Build 11929.20254)). Where possible, data are given as median and standard deviation for quantitative variables, and as a percentage for qualitative variables. Ninety-five percent confidence intervals were calculated when data where justifiable and possible. 

The primary outcome of interest was vancomycin concentration at time (t) while undergoing dialysis. Clearance was assumed to follow first order kinetics while undergoing dialysis; that is, clearance was assumed to be proportional to plasma concentration *C* (Equation (1)).
(1)$$\frac{dC}{dt}=-kC$$

It follows, therefore, that a linear relationship should exist between log concentration and time, Equations (1)–(5):
(2)$$\frac{1}{C}\cdot dC=-k\cdot dt$$
(3)$$\int \frac{1}{C}\;dC=\int -k\;dt$$
(4)log (C)=−kt+c
(5)log (C)−c=−kt

It follows that *c* should be the logarithm of *C*_0_, the starting plasma concentration:(6)$$\log\left(\frac{C}{Co}\right)=-kt$$

Fitting the linear relationship from Equation (6) was done by least squares regression, with log (*C*/*C*_0_) as the outcome and time the explanatory variable. 

The primary analysis was a determination of whether the slope of this line differed between the medium cut-off membrane dialysis (Theranova) and high-flux membrane dialysis (Revaclear), indicating different clearance of vancomycin. Multiple measurements were taken (at different times and on different days) from each individual patient. To increase statistical efficiency, all measurements were used to fit the lines. However, measurements from one individual are likely to be correlated, violating the assumption of independence used for most standard statistical techniques. Therefore, standard errors were calculated using multivariate methods developed for longitudinal data analysis that are able to account for within-individual correlation (namely, generalised estimating equations and multivariate bootstrapping). Elimination half-life (T_1/2_) was calculated for each membrane. 

The vancomycin concentration reduction from blood during haemodialysis sessions were calculated from the following formula and expressed as % concentration reduction: % vancomycin reduction = [(C_preHD_ − C_120minHD_)/C_preHD_] × 100 where C_preHD_ = The concentration of vancomycin before haemodialysis, C_120minHD_ = the concentration of vancomycin at 120 min into dialysis.

Other analyses involved examining the effects of BMI, dialyser membrane, blood flow (Qb), different vancomycin trough concentration, use of loop diuretics, use of Angiotensin Converting Enzyme inhibitor (ACEi)/Angiotensin Receptor Blocker (ARB), interval between dialysis session, interval between doses of vancomycin delivered and duration of haemodialysis session. For these analyses, only percentage reduction of vancomycin plasma concentration at 120 min of dialysis was used as the outcome. Generalised estimating equations were used to account for within-individual correlation.

No formal power or sample size calculations were made. The data collected was used to estimate possible effect size and variability of measures and allow us to assess drug clearance. A total of five volunteers on HD presenting with an acute infection/indication requiring vancomycin participated in the study. This allowed us a maximum of 30 sessions to obtain data. In general, all the analyses described are considered exploratory and are designed to give some insight into the relationship between the treatment and potential outcomes.

## Figures and Tables

**Figure 1 toxins-12-00317-f001:**
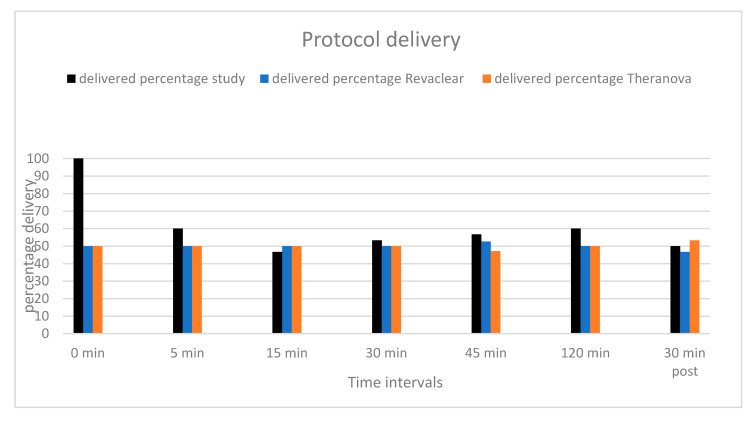
Protocol delivery by dialyser.

**Figure 2 toxins-12-00317-f002:**
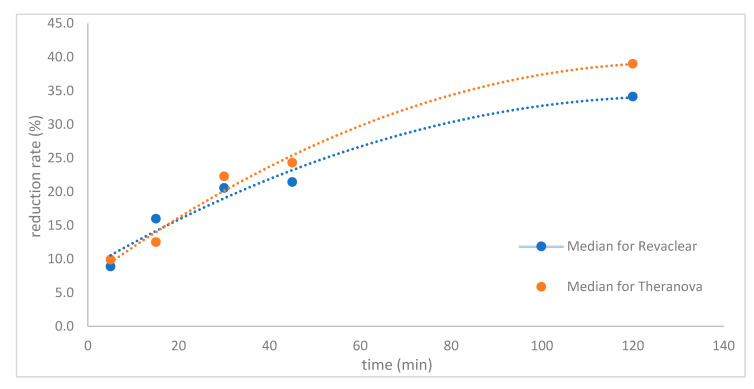
Vancomycin median percentage reduction with time.

**Figure 3 toxins-12-00317-f003:**
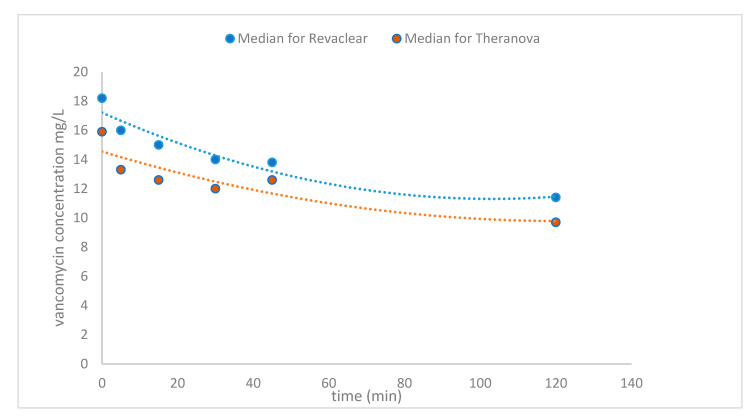
Vancomycin concentration reduction with time.

**Figure 4 toxins-12-00317-f004:**
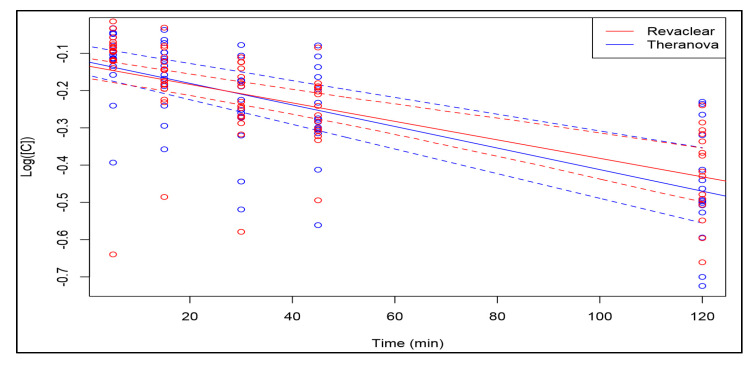
Logarithms of (C/C_0_) as a function of time, with least-squares linear estimates (solid lines) and the associated 95% confidence intervals (dashed lines) shown.

**Figure 5 toxins-12-00317-f005:**
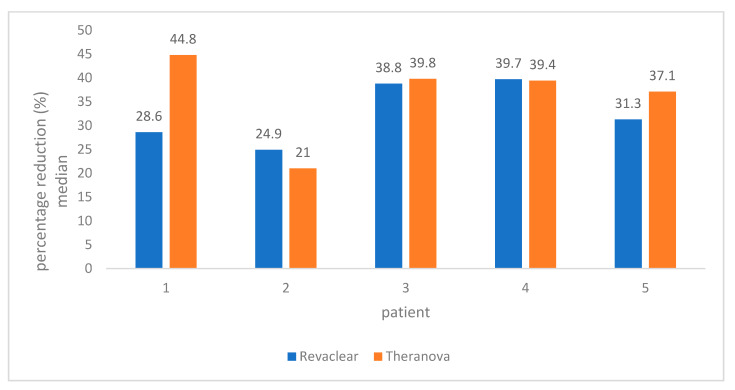
Median percentage of vancomycin removal at 120 min per patient.

**Table 1 toxins-12-00317-t001:** Patients’ demographic and clinical characteristics.

Characteristic	Number of Patient (%)
Sex	
Female	4 (80)
Age	
≤65	3 (60)
Ethnicity	
Maori	4 (80)
Caucasian	1 (20)
BMI	
≤30	2 (40)
Access used	
Fistula/Graft	2 (40)
Central lines tunneled or temporary	5 (60)
Bloods Flow (pump velocity) mL/min, per session	
≤250	16 sessions (53.3)
>250	14 sessions (46.7)
Use of loop diuretics	3 (60)
Use of ACEi or ARB	1 (20)
Patients with vancomycin trough ≥15 mg/L on day 1	3 (60)
Number of iHD per week = 3	5 (100)
Duration of HD per session	
≤240min	1 (20)
>240min	4 (80)

BMI = Body Mass Index (kg/m^2^), ACEi = Angiotensin Converting Enzyme Inhibitors, ARB = Angiotensin Receptor Blocker, iHD = Intermittent Haemodialysis, HD = Haemodialysis.

**Table 2 toxins-12-00317-t002:** Patients’ indication for treatment with vancomycin.

Patient	Indication for Treatment
1	Dialysis Tunnelled central line sepsis
2	Peritoneal Dialysis catheter exit site infection with MRSA
3	Dialysis Tunnelled central line sepsis
4	Coagulase negative staphylococcus species bacteraemia
5	Cellulitis/Infected synthetic Arteriovenous Graft

MRSA = methicillin-resistant staphylococcus aureus.

**Table 3 toxins-12-00317-t003:** Patients’ individual variables and effect on vancomycin percentage reduction at 120 min.

Patient Variable		Median Study Percentage Reduction (%)	Median High-Flux Membrane Percentage Reduction (%)	Median Medium Cut-Off Membrane Percentage Reduction (%)
BMI (kg/m^2^)	<30	39.3	36.5	42.5
	≥30	34.4	31.3	37.1
Qb (mL/min)	≤250	31.3	31.3	31.3
	>250	38.5	34.9	39.2
Vancomycin concentration at start of HD (mg/L)	<15	33.8	32.7	33.8
	≥15	38.8	34.9	39.3
Vancomycin concentration at start of HD (mg/L)	<18	34	31	38
	≥18	39	38.8	39.2
Loop diuretics	Yes	38.5	34.9	39.2
	No	30.6	30.6	29.4
ACEi/ARB	Yes	36.7	28.6	44.8
	No	36.4	34.5	38
Interval between HD session (Hours)	≤48	37.6	34.9	39.2
	>48 (i.e., 72hs)	34.9	32.7	37.3
Interval between doses (Hours)	≤48	36	34.9	37.1
	>48 (i.e., 72hs)	37.2	32.7	39.2
Duration on HD (minutes)	≤240	37.1	31.3	38
	>240	35.6	34.5	39.2

BMI = Body Mass Index, Qb = blood flow on dialysis (pump velocity), ACEi = Angiotensin Converting Enzyme Inhibitors, ARB = Angiotensin Receptor Blocker, HD = Haemodialysis.

**Table 4 toxins-12-00317-t004:** Secondary analysis for the effects of loop diuretics, blood flow on dialysis and BMI on the vancomycin level at 120 min of dialysis.

Variable	Estimate	Concentration Ratio
(Intercept or Baseline)	−0.43	NA
Medium cut-off membrane	−0.04	0.96
Use of loop diuretic	0.05	1.05
Blood flow of dialysis (per mL/min)	−0.001	1.00
BMI (per kg/m^2^)	0.01	1.01

**Table 5 toxins-12-00317-t005:** Vancomycin dose and infusion duration.

Loading Dose	Infusion Duration
750 mg	60 min
1000 mg	60 min
1250 mg	90 min
1500 mg	90 min
1750 mg	120 min
2000 mg	120 min

Pre-HD = pre haemodialysis, con = concentration (mg/L), % = percentage, min = minutes, MCO = medium cut-off membrane, HF = high-flux.

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
