# Peer review of "A Pharmacokinetic Study Comparing the Clearance of Vancomycin during Haemodialysis Using Medium Cut-Off Membrane (Theranova) and High-Flux Membranes (Revaclear)"

_toxins, 2020, doi:10.3390/toxins12050317_

Round 1
Reviewer 1 Report
The manuscript is very interesting, considering the high incidence of sepsis, in particular in dialysis patients and in intensive therapy, where also AKI is very frequent.
Furthermore, the drug resistance of antibiotic therapy is certainly a very current problem, which can create many difficulties, especially for dialysis patients, who are immunosuppressed and in whom it can be difficult to evaluate the correct blood concentration due to extracorporeal therapy.
However it is not very original:
-Removal of vancomycin administered during dialysis by a high-flux dialyzer. Nyman HA et al. Hemodial Int. (2018)
-Vancomycin during the Last Hour of the Hemodialysis Session: A Pharmacokinetic Analysis. Ghouti-Terki L et al. Nephron. (2017)
-Vancomycin dosing in chronic high-flux haemodialysis: a systematic review. Hui K et al. Int J Antimicrob Agents. (2018)
Also these filters are already studied:
-Macías N, Vega A, Abad S, Aragoncillo I, García-Prieto AM, Santos A, Torres E, Luño J. Middle molecule elimination in expanded haemodialysis: only convective transport? Clin Kidney J. 2018 Dec 15;12(3):447-455.
-Misra M, Moore H.A clinical study comparing the basic performance and blood compatibility characteristics of Nipro ELISIO-H®, Gambro Polyflux Revaclear® , and Freseniu s Optiflux® dialyzers. Hemodial Int. 2018 Oct;22(S2):S15-S23.
even the conclusions reached by the authors ………”Application of vancomycin during the last one to two hours of each dialysis is required to maintain therapeutic concentrations to minimise loss through the dialyser and maintain therapeutic levels”…….
are already used in clinical practice and confirm other studies in the literature
-Vancomycin during the Last Hour of the Hemodialysis Session: A Pharmacokinetic Analysis. Ghouti-Terki L et al. Nephron. (2017)
The authors reported….. “Vancomycin is widely used in HD patients for treating infections of vascular access”, surely but not only, this clarification in the abstract seems reductive and useless.
…………….Three patients were presumed to have residual renal function as they were on high dose loop diuretics………………….
it would be interesting to know how long these patients carried out hemodialysis, also because it is difficult, unlike peritoneal dialysis, to maintain a residual diuresis for a long time
In the indications to carry out vancomycin is also reported: …………Peritoneal Dialysis catheter exit site infection with MRSA…., it is not clear why a patient on hemodialysis has a peritoneal catheter, a source of infection?
The authors should better explain what their paper adds to the scientific literature.
I find this recommendation useful and necessary………………………”Pre-HD vancomycin serum concentration monitoring is recommended with dosage adjustment based on measured vancomycin trough concentrations”…………………The sample is certainly very limited
The manuscript should be revised by a na
tive speaker english
There are some typos
Author Response
Thank you for taking the time to provide your detailed review of our work. Please find below a point by point response to your suggestions.
Point 1: However it is not very original.
Response:
As you have pointed out the last few years have seen a number of publications on the pharmacokinetics of vancomycin use in high-flux HD, nearly 20 years after high-flux HD has become standard of care for many units internationally. However, the new MCO dialysers have only be in clinical practice for a couple of years and this is the first clinical study to evaluate if the new membrane characteristics which provide a much higher molecular weight cut-off and unique sieving co-efficient change the pharmacokinetics of this commonly used antibiotic.
Point 2: Also these filters are already studied
even the conclusions reached by the authors ………”Application of vancomycin during the last one to two hours of each dialysis is required to maintain therapeutic concentrations to minimise loss through the dialyser and maintain therapeutic levels”…….
are already used in clinical practice and confirm other studies in the literature
Response:
The study of the MCO dialyers use in clinical practice is very limited so far. While there have been some studies evaluating the safety and efficacy of these membranes, this is the first study to evaluate drug pharmacokinetics during an HD treatment using these membranes.
Point 3: The authors reported….. “Vancomycin is widely used in HD patients for treating infections of vascular access”, surely but not only, this clarification in the abstract seems reductive and useless.
Response:
This has been adjusted
Manuscript Line 10: Vancomycin is widely used in HD patients for treating a variety of infections.
Point 4: Three patients were presumed to have residual renal function as they were on high dose loop diuretics it would be interesting to know how long these patients carried out hemodialysis, also because it is difficult, unlike peritoneal dialysis, to maintain a residual diuresis for a long time
Response:
We have added this information to the results section:
“Duration of dialysis vintage was from 9 months to 4 years.”
Point 4:
In the indications to carry out vancomycin is also reported: …………Peritoneal Dialysis catheter exit site infection with MRSA…., it is not clear why a patient on hemodialysis has a peritoneal catheter, a source of infection?
Response:
This patient haemodialysis access was becoming very difficult and they were in the process of being transitioned to PD.
Point 5:
The authors should better explain what their paper adds to the scientific literature.
I find this recommendation useful and necessary………………………”Pre-HD vancomycin serum concentration monitoring is recommended with dosage adjustment based on measured vancomycin trough concentrations”…………………The sample is certainly very limited
Response:
We have revised the conclusion to make it clearer what this paper adds to the literature:
“While there are a wide range of recommendations regarding dosing of vancomycin in patients on high-flux HD, this is the first study to evaluate the pharmacokinetics of vancomycin clearance on the novel MCO dialysers. We identified vancomycin removal to be higher on MCO dialysis compared with high-flux HD with a predicted shortening of serum half-life to 240 minutes from 280 minutes. The clearance of vancomycin on HD is already significantly altered by patient characteristics and the dialysis script, therefore the addition of the MCO membrane provides another variable to the equation. We therefore recommend dosing of vancomycin is based on the pre-HD trough levels on an individual basis to avoid sub-therapeutic treatment.”
Point 6:
The manuscript should be revised by a native speaker English
Response:
The authors are both native English speakers and have re-reviewed the manuscript to pick up any typos. Thank you for highlighting this concern.
Reviewer 2 Report
While the premise is highly interesting, the small sample size of 5 patients I beleive is problematic. The most difficult thing to ignore in my opinion, is how do we know the different values observed are significantly different, a biostatistical consultation to see how the different curves can be compared would be advisable to ensure that we are presenting a statistically significant difference, I also beleive a larger study (10 patients or so) with an improved mix of male to female patients (who have different Volumes of Distribution) would be advisable.
Author Response
Thank you for taking the time to review our manuscript. Your suggestions resound with our own reflections on the study.
While designing this study as the senior investigator, I was also Chief Investigator to the REMOVAL-HD study a multi-centre study addressing the safety and efficacy of the MCO-membranes in a chronic HD population. So to organise larger studies to address the utility of the MCO membranes has not been a concern to me.
However, choosing a smaller population for this kinetics work was based on previous kinetic studies we have undertaken and unfortunately this time we found our sample size was not large enough to reach statistical significance. Although we have demonstrated a clear trend towards the increased clearance and reduced half-life we anticipated to see.
We will now go back to our ethics board and ask for an extension of the study to enable further recruitment. Assuming this is granted we will be looking at least a 12 month period before this manuscript could be updated. During that time the MCO dialysers are being used internationally in routine clinical care and it is therefore important for nephrologist to be aware of our findings even while the study is expanded i.e. a report of preliminary findings.
Many thanks
Round 2
Reviewer 1 Report
The authors have made the requested changes, therefore the manuscript can be accepted in its current form.
Reviewer 2 Report
the manuscript has been sufficiently improved